# Comparative Analysis between the Role of Local Communities in Regional Development inside Japanese and Russian UNESCO's Biosphere Reserves: Case Studies of Mount Hakusan and Katunskiy Biosphere Reserves

Aida Mammadova [1,*] , Christopher D. Smith [2] and Tatiana Yashina [3]

1    Organization of Global Affairs, Kanazawa University, Kanazawa 921-1192, Japan
2    Smith Custom Editing, Kanazawa 920-1156, Japan; chris@smithenglish.jp
3    Department of Protection, Katunskiy Biosphere Reserve, 649495 Ust-Koksa, Russia; altai-yashina@yandex.ru
*    Correspondence: mammadova@staff.kanazawa-u.ac.jp

**Abstract:** The United Nations Educational Scientific and Cultural Organization has designated the Man and Biosphere Program to foster a better relationship between the environment and people. The topic of this study is to elucidate the role of local communities in the regional development of Biosphere Reserves with a focus on management roles (top-down or participatory) and the motivational drivers of the people involved (ecocentric or anthropocentric). Based on qualitative interviews taken from the two case studies of the Mount Hakusan Biosphere Reserve in Japan and the Katunskiy Biosphere Reserve in Russia, a comparative analysis was conducted to explore the differences between the engagement of locals in the management of their biosphere reserves. This analysis examined relationships between the government and the local communities, the attitudes of the locals towards the biosphere reserves, and the historical perception on nature protection for each community. The findings showed that Russian biosphere reserves are mainly managed by local people who live inside the protected area while Japanese biosphere reserves are governed by local authorities and administration offices. This allows the Russian communities to have greater access to management processes, and therefore play a larger role in regional development.

**Keywords:** biosphere reserves; communities; governance; regional development; sustainability

## 1. Introduction

In 1971, UNESCO launched the Man and Biosphere Program (MAB) with the aim to build a sustainable relationship between people and their environments established on a scientific basis. The MAB designated the first Biosphere Reserves (BRs) in 1976 in eight countries (57 in number) as protected areas of terrestrial, coastal, and marine ecosystems established by the national government [1,2]. BRs are recognized as sites where human–nature interactions are tested, refined, and demonstrated and where the objectives of the MAB program are implemented. To strengthen its global network, the MAB program launched the World Network of Biosphere Reserves (WNBR), and by the year 2020, the number of BRs reached 714 in 129 countries [3]. To guide the WNBR, the MAB Strategy 2015–2025 along with the detailed Lima Action Plan 2016–2025 (LAP) were adopted to effectively implement the objectives of the program [4]. Each of the MAB National Committees of the WNBR are strongly encouraged to address and implement the LAP at regional, national, and international levels. The purpose of the MAB Strategy is to support Member States and stakeholders with four main objectives: (1) Conserve biodiversity, restore and enhance ecosystem services, and foster the sustainable use of natural resources; (2) Contribute to building sustainable, healthy and equitable societies, economies and thriving human settlements in harmony with the biosphere; (3) Facilitate biodiversity and sustainability science education for sustainable development (ESD) and capacity building; (4) Support



mitigation and adaptation for climate change and other aspects of global environmental change [5]. The key goal of the MAB strategy is to ensure that its World Network of Biosphere Reserves effectively collaborate and network with the members of the WNBR to develop and strengthen models for sustainable development. It is important to create a dialog between members of the WNBR to communicate experiences and lessons learned. This communication encourages information sharing on good practices and enables the global dissemination and implementation of these models.

The WNBR promotes the integration of people and nature through a participatory dialog about sustainable development. The role of local communities within the regional network is indispensable in achieving the UN's 2030 Agenda along with the Sustainable Development Goals (SDGs) [6]. The local involvement plays a key role in sustainable ecosystem management and the development of the community capacity [7] as well as being mentioned in the MAB Strategy and the LAP. However, until now, protected areas were focused mainly on biodiversity monitoring to prevent biodiversity loss, overlooking the needs of the local communities and populations [8]. The demand for scientific and data-based knowledge played the main role for ecosystem management [9]. When they were created, most of the BRs were designated within existing protected areas [10], and they were focused on the protection of natural and genetic diversity, promoting environmental research and educational activities [11]. This created a severe concern for the role of local communities who were the main actors in sustainable natural resource utilization and management of protected areas [12]. However, with the adoption of the Seville Strategy and the Statutory Framework in 1995 [13], a new policy was initiated into the protected areas. This policy integrated the interaction between the needs of local people and nature. The Seville Strategy emphasized including appropriate zonation around the protected areas to promote socio-ecological interactions between humans and nature. Since that time, UNESCO BRs have consisted of three zones: a transition zone, a buffer zone, and a core zone [14]. These three zones are associated with different degrees of natural resource use and anthropogenic influence. Presently, BRs are considered to be ecological model regions with the local people playing an integral role in the successful implementation of the reserve concept.

Communities play an essential role as the keepers of regional biocultural diversity and identity. More research must be conducted to evaluate a community's capacity for managing BRs. Local people, in most of the cases, perceive protected territories as restrictions for action, rather than opportunities to increase socio-economic development. To evaluate the role of the community, we need to first understand the structure of how communities are built inside BRs and what creates the obstacles for their sustainable development. Community studies have been conducted by several foreign researchers by assessing the well-being of communities [15], resilience [16], sustainability [17,18] and quality of life [19]. Building strong communities has been a recent trend in the WNBR, and UNESCO has emphasized the effectiveness of BR management with community participation [20]. Some overseas countries provide National Community Capacity Building Programs to achieve community empowerment, such as National Standards for Community Engagement [21]. Those countries even provide community capacity building tools [22] and contribute to the creation of national partnerships inside BRs through community building [23]. Well-organized communities provide positive environmental conditions, and therefore attract more people to the region. Understanding the community's capacity in ecosystem management can help rural areas to deal with depopulation problems and attract younger people by providing job opportunities inside the region using natural resources. Some examples are the promotion of eco- and green-tourism, local and safe food production, and craft-making using forest resources.

## 2. Conceptual Framework of Community Participation and the Objectives of This Study

Local communities are "residents living near (or sometimes within) a protected area" [24]. UNESCO requires that biosphere reserves are managed and planned through

participation, involvement, and engagement. Furthermore, the effective management of BRs relies directly on how well those communities are engaged and participate in the management processes of the protected areas. The participation of local communities is defined as "engagement processes and practices in which a wide range of people work together to achieve a shared goal guided by a commitment to a common set of values, principles and criteria" [25]. Past studies have already shown the importance of participation by locals in BR management, and several of these were utilized to define and examine the roles of participation for our study. The first is a framework consisting of 11 major factors which influence BR management by Van Cuong et al. [26]. In their studies, which included ninety BRs with sixty successful and thirty less successful BRs, they ranked these 11 factors from the highest (1) to the lowest (11) (Table 1). "Participation" with the elements of engagement and collaboration of local communities was the most important factor that influenced the success of BRs.

**Table 1.** Major factors influencing biosphere reserve management. Adapted from Van Cuong et al. [2].

| | Factors | Description |
|---|---|---|
| 1. | Participation and collaboration | Participation, engagement, collaboration of local community, public, private stakeholders, NGOs |
| 2. | Governance | Leadership, coordinating agency, building partnerships, government and stakeholder commitment, support and on-going support |
| 3. | Awareness and communication | Understanding BR concept and MAB program, liaison, communication program, stakeholders have a sense of BR ownership |
| 4. | Landscape and zonation | Application of landscape and zonation to fulfil all 3 desired functions across different land uses |
| 5. | Regional integration | Link to regional development, socio-economic program and other management systems in the region |
| 6. | Learning orientation and system thinking | Use of BR as living laboratory, experiment application, adaptive management, learning by doing |
| 7. | Finance and resources | State funding availability, support projects and human resources (number, quality, education, professional experienced staff) |
| 8. | Economic development | Economic development, livelihood and production, tourism development, branding and marketing activities |
| 9. | Management and implementation | Management plans and vision, ground activity implementation, law enforcement |
| 10. | Monitoring and evaluation | M and E frequency, measurement of tangible indicators |
| 11. | Research linkage | Partnership with research institutes, universities in research |

Similarly, Stroll-Kleemann and Welp [27], in their studies in Finland, Estonia and Germany, identified fifteen factors for the successful management of BRs. They showed that the factors dealing with participation were ranked second (collaboration with local authorities) and sixth (community participation). Ferreira et al. [28] developed a conceptual framework with four categories for successful BRs management. The framework highlights four main categories, one of which is a "process" category that includes management and governance. It is also associated with decision-making. In this category they highlighted the importance of community participation for successful BR management as "participatory processes". Several studies have also shown the importance of public participation [29]; however, there is still a significant gap between successful community engagement using a participatory approach [30] and conventional top-down management [31] which is still used in many BRs. In 2008, the Madrid Action Plan included new targets along with a

series of actions related to the participation of communities in BRs [20] with a participatory bottom-up approach to improve environmental, social, and economic sustainability.

The term "participation approach" can be defined in various ways. Diamond et al. [32] described two types of participatory approaches: the first by "means", where local people cooperate or collaborate with the external institutions, and this participation becomes the means for effective implementation; the second is by an "end", where participation is viewed as a goal and refers to empowering people with particular skills, knowledge, and experience that is needed for their own development. Moreover, different levels of participation exist from simple information sharing to power transfer and decision-making authority. The second, and the most essential framework used in this study is drawn from an explanation of participation described by Borrini-Feyerabend [33], Pimbert and Pretty [34], Diamond et al. [32], and Agarwal [35]. In this framework, at least seven levels of participation have been outlined, according to the degree of involvement (Table 2).

**Table 2.** Different levels of participation. Adapted from Borrini-Feyerabend [33], Pimbert and Pretty [34], and Agarwal [35].

| Form/Level of Participation | Characteristic Features |
| --- | --- |
| Level A—Nominal participation | Membership in the group, when no interaction occurs between local stakeholders and managing institutions |
| Level B—Passive participation | Being informed of decisions ex post facto, when the project already began; or solely attending the meetings without the right to speak up |
| Level C—Consultative participation | Being asked for the specific opinion, but without guaranteeing of influencing decision |
| Level D—Activity-specific participation | Being asked (or volunteering) to undertake specific tasks |
| Level E—Active participation | Expressing opinions, whether or not solicited, or taking initiatives of other sorts and projects |
| Level F—Interactive (empowering) participation | Having a voice and influence in the group decisions |
| Level G—Self-mobilization and taking over of the responsibility | Local people assume primary management responsibility, by taking initiative independently from the external institutions |

According to the Diamon et al. description [32], participation is seen as "means", or method, within levels A, B, C, and D, and will lead to no action by local communities. This approach is very often seen in top-down models of the designation of protected areas. In these top-down cases, authorities and governments apply conservation policies and strategies without consultation with local people. The E, F, and G levels correspond to bottom-up models. In these three cases, participation is an "end" to empower local communities to be actively involved in decision-making processes. However, even though levels E and F can still be conducted together with higher authorities, level G provides complete independence from external institutions. Brechin and West [36] argue the necessity of linking top-down and bottom-up approaches to achieve sustainable conservation. Indeed, in the long term, a bottom-up approach alone would not work without the necessary strategy and involvement of diverse stakeholders along with financial support.

Other than the nexus of these two approaches, community participation also strongly depends on how people relate to and perceive the protected areas and natural environment. McNelly [37] proposed that local communities would support conservation initiatives if they received direct benefits from the initiatives that could meet their needs. Therefore, how well local communities perceive and evaluate the protected area would induce participation in the conservation initiates. Wallner et al. [38] identified three key categories which influence local residents' perceptions and evaluations of BRs, i.e., the economic situation, the history of nature protection, and the power balance between the involved stakeholders. The evaluation of BRs will increase in locals if they have any possibility to gain some income, economic benefit, or employment opportunities [39]. Hernes and Metzger [40] argued that evaluating the local residents' environmental values, worldviews, and attitudes would help understand people's conservational behaviors. Linking attitudes with behavior, Gagnon-Thompson and Barton [41] have suggested two motives that underline the environmental

actions in locals. One is "ecocentric", where individuals value nature for its own sake and protect nature for its intrinsic values. The other is "anthropocentric", where individuals feel that nature should be protected for the positive effects that enhance the quality of life for humans. Identifying historical backgrounds, motivations, economic values, and perceptions about protected areas can provide clarity about the willingness of locals to contribute to the effective management of BRs.

As described above, there were two primary frameworks utilized in this study. The first was the framework by Van Cuong et al. [26] (Table 1) which was used to analyze the participatory approach. The second was the framework by Borrini-Feyerabend [33], Pimbert and Pretty [34], Agarwal [35] (Table 2) which was used for perception analysis. The objectives of this study were: (1) to contribute to studies of the perceptions and attitudes of people towards protected areas for the successful management of BRs and (2). to conduct a comparative analysis examining the role of local communities in regional development inside Russian and Japanese BRs. Our hypothesis was that the perception of protected areas depends on the historical relationship between local governments and communities, the attitudes and economic perspectives of locals, and finally, the cultural perception of nature protection. According to our objectives, we have examined the following questions:

1.  What are the differences in the relationships between local communities and the government concerning the management of BRs between both countries? Are those relationships built on top-down or participatory approaches?
2.  What are the main motivational differences of the concept of nature protection between both countries? Are those motivations based on ecocentric or anthropocentric drivers?

Although Russia and Japan, as neighboring countries, share a similar natural environment and have deeply rooted cultural traditions regarding the appreciation of nature [42], they have differences in historical, social, and economic aspects of nature management. The Japanese history of BRs is strongly related to National Parks [43], however, BRs and National Parks are governed by different Ministries. National Parks belong to the Ministry of Environment (MoE), whereas BRs are regulated by the Ministry of Education, Culture, Sports, Science and Technology (MEXT). Presently, Japan has 10 BRs, and mainly all Japanese BRs are designated within existing National Parks. In Russia, the idea of nature conservation is deeply rooted in history, and there are very strict, state-run nature protection regulations for areas called "Zapovedniks" (Strict Scientific Nature Reserves) [44]. The idea of "Zapovednik" is different from national parks and excludes any human activity except the activities of rangers and scientists which also prohibits economic activities. Until 2000, all BRs were nominated only from Zapovedniks, with limited community activities. It was not until 2001 the status of BRs was nominated for the National Parks to increase tourism and recreational activities. At present, the Russia Federation has 47 BRs with only one transboundary BR with the Republic of Kazakhstan [45], 101 Zapovedniks with 35 of them having the status of BRs, and 48 national parks with 7 having the status of BRs (Social Information Agency). Zapovedniks and BRs are both regulated by the Ministry of Natural Resources and Environment of Russian Federation.

Being part of the MAB Program, both countries can contribute to regional development through the engagement of local communities. As a part of the WNBR, they can share lessons learned for the better governance and management of BRs. Japan and Russia are included in the East Asia Biosphere Reserves Network of the WNBR. Japan neighbors, Russia's Far East and Siberian regions, share common biodiversity and bioresources. In 2001, the first workshop was held on the transboundary cooperation of biodiversity conservation between the Russian and Japanese BRs, especially related to Kunashir, Irurup, Shikotan, and the Habomai Islands at the National Olympic Youth Center in Tokyo, during the UNESCO/MAB IUCN Workshop [46]. The designation of the transboundary BRs between Russian and Japanese territories is still in question. There is an important need to carefully analyze not only the biological and cultural diversity of the regions, but also the locals' livelihoods and the ecosystem management between the two countries. The

countries of Japan and Russia both need to evaluate the capacity of local communities in BR management to face regional problems. Until now, no studies have been done to compare Japanese and Russian BRs, and very little information exists concerning the relationship between governance and the engagement of local communities in the management of BRs in both countries. We believe that this kind of cross-country analysis can contribute to a better understanding of human aspects on the demographic outflow of the local residents from BRs. It can also promote solving mutual intercultural issues and help in further management of the neighboring territories concerning land protection and planning. In addition, the comparative analyses between both countries should contribute to the MAB Strategy and the LAP on effective collaboration within the regional members of WNBR.

## 3. Methods and Case Study Areas

The Japanese Mount Hakusan BR (hereafter as MHBR) and the Russian Katunskiy BR (hereafter as KatBR) (Figure 1) were chosen as the fields of our study, according to the similarities in the following criteria shown in the Table 3.

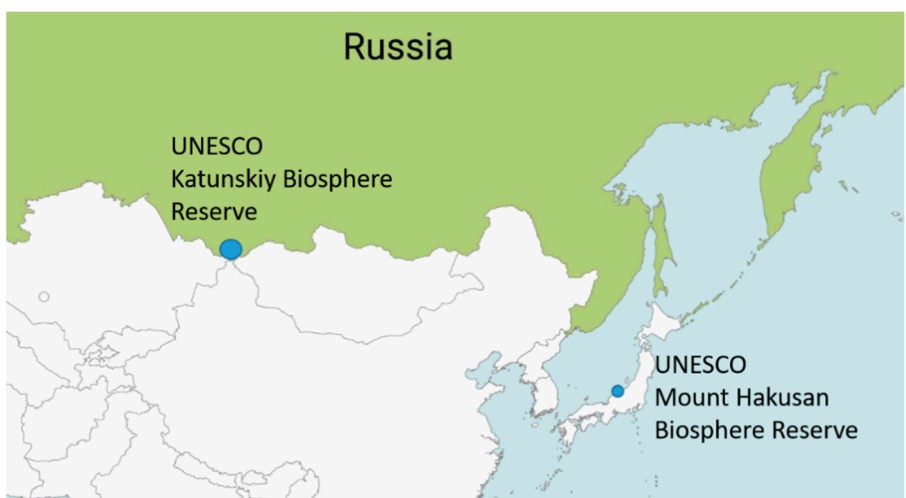

**Figure 1.** Overview of the locations of UNESCO Mount Hakusan Biosphere Reserve in Japan and Katunskiy Biosphere Reserve in Russian.

**Table 3.** Main similarities between MHBR and KatBR. "O" indicates the criteria was applicable for the region. "X" indicates "not applicable".

| Description | MHBR in Japan | KatBR in Russia |
|---|:---:|:---:|
| Human Settlement | | |
| • Mountainous regions | O | O |
| • Transition zone and periphery of the BRs | O | O |
| • Remote access away from cities | O | O |
| Livelihood | | |
| • Mountainous Believes and Culture | O | O |
| • Mountainous Tourism | O | O |
| • Apiculture | O | O |
| • Forestry | O | O |
| • Hunting | O | O |
| • Cattle and sheep farming | X | O |

The main criteria to select these two BRs was the remote location of the villages away from the surrounding big industries. The most important question was whether people living in those remote areas were using the concept of BRs to support their livelihoods. Even though the designation periods, economic situations, and planning processes are different between them, they have shown similarities in livelihoods, demographic situation, and geographical locations.

The research method included a literature review, qualitative interviews, and qualitative comparative analysis between the two BRs. According to our research goal to elucidate the relationship between the local communities and government, as well as participants' perceptions and motivations towards the BRs, we concluded that a qualitative case-study approach would be adequate. A qualitative approach allows the generalized understanding of values [47] and provides insight into the social reality of given circumstances [48], which can be applied to other cases with similar conditions and comparable issues. A qualitative approach also provides a deeper understanding of the particular phenomena and permits the informants to freely structure and define their responses [49]. The data from all stakeholders' groups was collected through semi-structured interviews with open-ended questions. Semi-structural interviews [50] were conducted either during personal meetings or via web-based telecommunication services with guaranteed anonymity for all respondents. Stakeholder groups for the interviews were selected after a primary literature review. To select the interviewees, the interviewer asked the reserve managers to visit each BR and contact the relevant stakeholders who could take part in the study. Snowball sampling was further used to identify additional respondents. This sampling included local people who were involved with conservation, tourism, community development issue, as well as local landowners. Later two groups of interviewers were selected that have impact on the management of BRs in both countries. These were: (1) directors and managers of BRs, i.e., founders, board members, staff, rangers, and volunteers and (2) local communities. i.e., residents who live inside the village and represent important activities such as agriculture, forestry, tourism, NGOs, and local business owners.

1. Directors and managers of BRs: For the Russian site, the primary interview was conducted with a representative of the Russian National Committee of the MAB Program, in March 2017. The interviews in KatBR were conducted with the vice-director, 2 local managers, and 2 inspectors, during a visit to the area in March 2017, in Altai, Russia. The interview with the Director of the KatBR was conducted in 2018, during the "15th Meeting of the East Asian Biosphere Reserve Network" in Kazakhstan, Almaty.

   For the Japanese site, interviews with the three local managers of the MHBR were conducted in 2020 during the "Mount Hakusan Biosphere Reserve Community Development Workshop" in Hakusan City, Japan. In total, 33 people took part in the workshop which included the youth working inside the MHBR, local representatives and organizations working in the region, people who have moved to the MHBR from other prefectures, and some other groups of people who had an interest in the MHBR. The workshop was designed to, first, identify the common questions, then, to identify issues in each region, and finally, to provide future recommendations. The most relevant issues and recommendations concerning the community participation were selected. Workshop participation provided important insights into local and institutional realities. Data collected from this workshop was used for the case study analysis.

2. Local Communities: Interviews were conducted in the village of Shiramine in MHBR, Japan, and Ust-Koksa village in KatBR, Russia. These two villages were selected because of their remote location in the transition zone, and periphery of each BR, and because the residents are mainly natives with no foreign settlements for many generations. Village locations are depicted in Figures 2 and 3. Interviews were conducting by visiting each village during the following periods: March 2018, Ust-

Koksa (11 people interviewed) and two visits to Shiramine in the summer 2019–2020 (11 people interviewed).

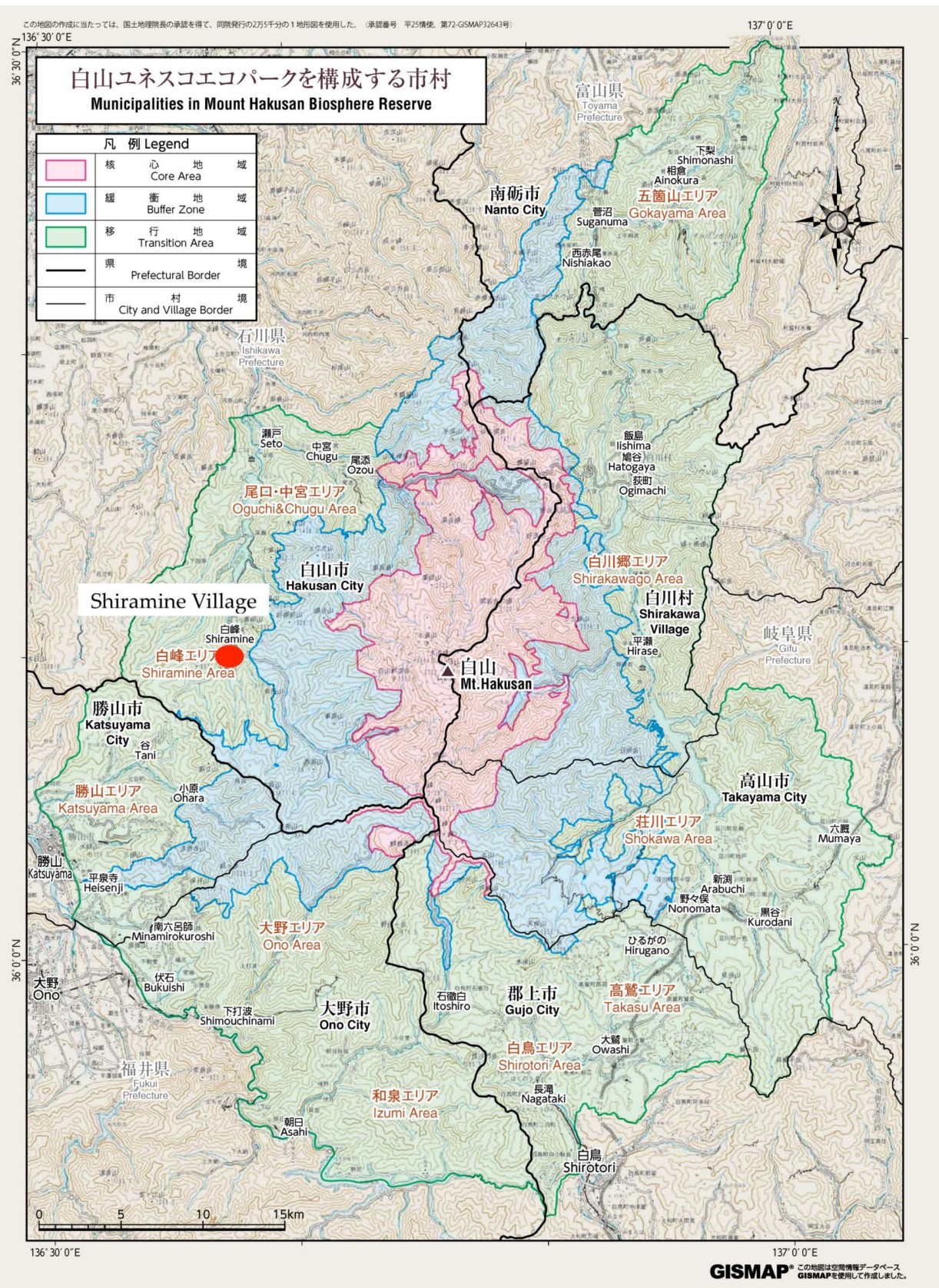

**Figure 2.** Territory of Mount Hakusan Biosphere Reserve, with municipalities and three zonations. Source: https://hakusan-br.jp/english/ ( accessed on 12 August 2021)

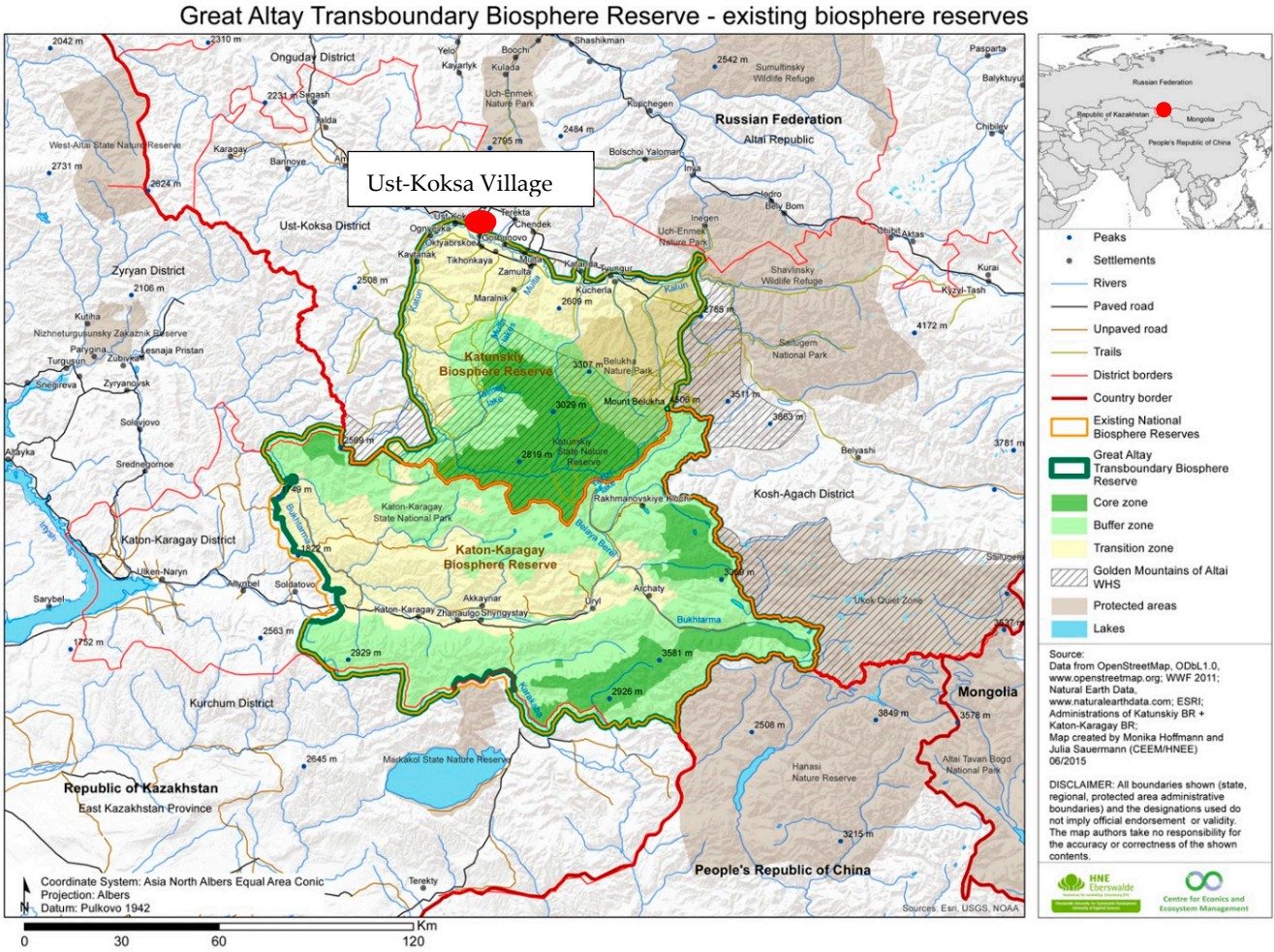

**Figure 3.** Transboundary territory of Katunskiy Biosphere Reserve (Russia) and Katon-Karagay Biosphere Reserves (Kazakhstan).

In total, 26 qualitative interviews were conducted (14 in MHBR and 12 in KatBR). The details of the interviewers are shown in the Table 4.

**Table 4.** Details of the interviewed personnel in MHBR and KatBR.

| Interview Groups and Places | Interviewed People | Total |
|---|---|---|
| MHBR in Japan<br>1. Directors and managers of BRs community development workshop | 3 Employers | 14 |
| 2. Local communities<br>Shiramine Village | 3 Local guides<br>1 Shop keeper<br>2 Elderly employers in the village<br>1 Community Leader<br>2 Local farmers<br>2 Young employers | |
| KatBR in Russia<br>1. Directors and managers of BR,<br>During the 15th meeting of EABRN and visiting Ust-Koksa Village | 1 Director<br>1 Vice-director | 12 |
| 2. Local Communities<br>Ust-Koksa Village | 2 Employers<br>2 Inspectors/rangers<br>3 School Teachers<br>1 Local craft man<br>1 Housewife<br>1 Unemployed | |

Interview questions were developed through the participatory observation of local communities and a literature review [51]. To test the questions, we conducted semi-structured interviews with two experts from MHBR (summer 2018) and one expert from KatBR (spring 2018). Regarding the objectives of the study and main research questions, the interview included four main parts with five questions each: (1) General information about BRs, (2) governance and management, (3) issues of the BRs, (4) engagement of local communities in BR management. Utilizing a strategy from Patterson and Williams [52], the interview began with general and broad questions about each BR to help interviewers feel comfortable and start a discussion. All interviews were taken individually either in Russian or Japanese with a duration of 40–60 min. These interviews were either audio-recorded or transcribed. In some cases, in order to make interviewers feel more relaxed and share more information, extensive notes were taken without recordings. Missing information and data from the interviews were supplemented by an analysis of annual reports, newsletters, and the websites of each BR.

The context analysis was conducted according to the thematic analysis approach based on the grounded theory method of Glaser and Strauss [53] by developing codes and generating a theory from the collected interview data [54]. To analyze the data, interview transcripts and notes were coded to generate topics that emerged from the interviews [55]. Interviews with authorities and local communities were coded separately in order to highlight the differences and similarities in perception and attitudes towards the BRs of each country. Questions which were related only to the directors and managers of BRs, i.e., question on "governance and management", were not asked to the local people, since they were not related to this section. Dominant topics were identified and the interrelation with other thematic categories were analyzed. Finally, based on the results, a perceptional framework was developed with three key components: (1) historical relationships between local governments and local communities, (2) attitudes and economic perspectives, and (3) perceptions of nature protection. Using this framework, a qualitative comparative analysis was conducted to elucidate the differences between both countries.

The analytic process was accompanied by a literature review of scientific papers, local reports, and publications. The literature and documentary reviews were conducted by focusing on the management systems of protected areas, the role of communities in promoting regional development in each BR, and reviewing the sources from the legislative and policy statements. Online research was conducted by accessing Wiley Online Library, Google Scholar, and Microsoft Academic Search to include relevant literature, primary sources, and related articles and books.

*3.1. Study Area: Japanese Mount Hakusan Biosphere Reserve*

The MHBR was one of the first designated BRs in Japan with a population of more than 17,000 and a total territory of 199,329 hectares. Agriculture and tourism are the main source of income. At its beginning, in 1962, the territory was recognized as "Hakusan National Park" and was shared between four Prefectures (Ishikawa, Toyama, Gifu, Fukui) [56]. In 1980, the territory of the national park (480 km$^2$) was also designated as the UNESCO Mt. Hakusan Biosphere Reserve, and the protected area of the national park was included in the core and buffer zones. This designation was conducted by local authorities and had no recognition among the local villagers. Because of the lack of recognition, it stayed dormant for more than 30 years without any practical municipal management. The revival of the MHBR started in 2010, when Japan was challenged by the MAB's "Seville Strategy" for Biosphere Reserves [14] declaring the expansion of the BRs to include the transition zones to promote regional sustainable development and induce the utilization of natural resources by the locals. The involvement of locals started in 2012, when a dialog was created between the governments of the four prefectures and seven municipalities, and a final agreement was made to apply for an extension of the transition zone. Figure 2 shows all three zones of the MHBR prefectures and municipalities. In 2014, the Mount Hakusan Biosphere Reserve Council was established as an official organization to conduct

the future management of the MHBR. It was composed of 12 members. The council also has an executive board with two working groups (WG) and an academic group [57].

To involve more local villagers in the management processes and to increase the awareness of the MHBR, the council started to conduct a series of symposiums and events surrounding the theme "rediscovering local values through the biosphere reserve". In total, from 2014 to 2015, seven municipalities in turn carried out the events for the locals by inviting diverse speakers. Finally, in 2016, through the recognition of the dedicated efforts of many people to protect and rationally utilize the natural resources of Mt. Hakusan, the extension and inclusion of the transition zone inside the MHBR was approved during the 28th Session of the International Coordinating Council of the MAB Program held in Lima, Peru [4].

One of the popular activities inside the Japanese BRs was to conduct educational activities and create logos. The MHBR council conducted a competition among local citizens to create a logo. In May 2017, the competition winner was announced together with the establishment of the logo.

### 3.2. Study Area: Russian Katunskiy Biosphere Reserve

The Katunskiy BR was established as a strict nature reserve (Zapovednik) in 1991. It is in the Central Altai Mountains on the border of Russia and Kazakhstan. The area of the reserve, which is now the core zone of the BR, is 151,000 hectares. In 1998, it was designated as the UNESCO World Heritage site "Golden Mountains of Altai". In 2000, more than 600,000 hectares of this protected area and adjoining lands with traditional land uses were designated as a UNESCO Biosphere Reserve.

In 2004, transboundary cooperation with Katon-Karagaiskiy National Park in Kazakhstan was initiated, resulting in the designation of the bilateral "Great Altai transboundary BR" in 2017. This was a first on the Asian continent [58] (see Figure 3).

This BR is in a remote rural area far from cities and industrial centers. The nearest city, Gorno-Altaisk, is 400 km away. Local communities rely on various types of agriculture (primarily cattle breeding and apiculture) as well as traditional land uses such as hunting and collecting of non-timber forest products for their livelihoods.

According to the principles of the World Network of Biosphere Reserves, the main activities of KatBR are related to the following:

- Protection of biodiversity and the environment;
- Research and monitoring, targeted at understanding climate change impacts on the area;
- Ecological education of local communities and visitors to the BR;
- Provision of sustainable livelihoods for local communities;
- Development of ecotourism.

Other than the departments of Protection, Research and Ecological Education, in 2015, KatBR created a public council to increase the involvement of local people in the management and decision-making processes of BR. The council was created as a platform to ensure the participation of all interested parties and local communities regarding the socio-economic development of territories in the transition zones. KatBR also has the Scientific and Technical Council, as an advisory body responsible for planning, management, and scientific and research activities [59].

Differences between the MHBR and KatBR in territorial area, population, governance, management, and organizational structure are shown in Table 5.

**Table 5.** Differences between Japanese MHBR and Russian KatBR.

| | MHBR in Japan | KatBR in Russia |
|---|---|---|
| Total Area | 199,329 ha | 587,949 ha |
| Zonation | Core: 22,120 ha, 11%<br>Buffer: 45,660 ha, 23%<br>Transition: 131,549 ha, 66% | Core: 151,664 ha, 26%<br>Buffer: 144,630 ha, 25%<br>Transition: 290,655 ha, 49% |
| Population | 17,000 pax | 4800 pax |
| Governance | Mount Hakusan BR Council: 4 Prefectures and 7 Municipalities | "Katunskiy State Nature BR" included inside "Great Altai Transboundary BR", together with Katon-Karagay BR in Kazakhstan has 9 villages |
| BRs management and engagement of locals | "Top-down" approach with regional management by the MHBR Council. Strategies developed by the authorities. | Participatory approach with initiative from local residents and interactive participation Develops strategies together with locals |
| Organizational structure and purposes | - Academic Board<br>- (responsible for scientific research)<br>- Executive Board<br>- (responsible for core, buffer zone management, and activities conducted between the local stakeholders inside the transition zone and local government)<br>- Administration | - Department of Protection<br>- Department of Research<br>- Department of Eco-education<br>- Public Council (responsible for cooperation with locals)<br>- Scientific and Technical Council (responsible for management, planning, protection and research) |
| Related Ministries | Ministry of Education, Culture, Sports, Science and Technology | Ministry of Natural Resources and Environment |

## 4. Findings and Analysis

We found several differences between Russian and Japanese BRs concerning biodiversity conservation, the nature management, and the locals' livelihoods. The details are shown in Table 6. According to the interviewees, Japanese BRs are facing problems with aging and depopulation due to a lack of education and community involvement in BRs. Dissimilarly, Russian interviewees identified problems with illegal poaching, biodiversity loss, and the management of large territories [60].

**Table 6.** Issues inside the MHBR and KatBR. Issues which were mostly mentioned by the interviewers (number of people >10) are indicated as significant issues "S". Issues which were less mentioned (number of people <10) are indicated as not significant "NS". Issues which were not applicable are indicated as "NA".

| Regional Issue | MHBR in Japan | KatBR in Russia |
|---|---|---|
| • Population outflow | S | S |
| • Lack of young followers | S | NS |
| • Less job opportunities inside BR | S | S |
| • Illegal poaching | NA | S |
| • Biodiversity loss | S | S |
| • Management of large territories | NS | S |
| • Constant change of working personal | S | NS |
| • Multi-governance issues | S | NS |

The Russian BRs are mainly managed by the local people who live inside the protected area, whereas Japanese BRs are governed by local authorities, administration offices and working personal is rotated every 2–3 years [61]. In Japanese BRs, local communities have little access to the management processes, and most of the time, they lack access to the information related to the BRs. In Russian BRs, the UNESCO designation provides more job opportunities for the locals.

The different participatory approaches in the management of BRs also showed significant differences. In the case of Russian BRs, until now, the concept of BRs was mainly scientific preservation and the ecosystem conservation. Less attention was given to the role of communities. However, from the 1990s, local people became more involved in management and decision-making processes. In the Japanese case, communities were not even informed about the designation of BRs in the UNESCO programs, and all activities were conducted by the local authorities without the participation of the local people.

*4.1. The Role of Local Communities in MHBR*

Until 1950, the entire area around Mt. Hakusan was the main supplier of silk, linen, industrial hemp, gun powder, medical herbs, firewood, and charcoal. However, after 1970 during a period of high economic growth and the rapid industrialization of Japan, an economical shift dramatically changed the socio-economic situation in Hakusan. Many job opportunities were lost, and new road construction into the mountainous regions triggered the local population to migrate from the mountain area into the cities. Statement by an elderly employer, "Road construction provided fast access into the cities, and many local people started to work outside. They moved their entire families to the urban areas so that their kids could get a better education and quality of life". Due to the population decline and outflow, the mountainous village landscapes rapidly changed. The ecosystem services were threatened to be lost. All four prefectures around Mt. Hakusan had the same issues of land abandonment and depopulation. Nowadays, the remaining population is elderly with only a few young followers. To deal with those issues, even during the dormant period of MHBR, residents were actively engaged in conservation activities, research, and education programs. They carried out many actions to pass on traditional culture to the younger generations. Unfortunately, those initiatives were not long-lasting due to a lack of successors.

A community leader stated, "One of the biggest issues in the village is old age and population decline. Each year we have around 10–15 people who pass away, and only 1–2 newborn children. If there are no people, then there is no future for the village and no future for the BR. So, how can we call this region sustainable?".

The revival of the MHBR by including the transition zone in 2016 brought a new hope to this region. It allowed the local residents to work together with the local government towards regional issues. However, there was a lack of awareness of the concept of BRs, and how the designation could contribute to regional development. A local villager stated, "For me, I know that Mt.Hakusan is a National Park, and that its territory is protected by the government, but I have never heard about the concept of UNESCO BRs and what its initiatives are. We have been living in these territories for many years, and nothing changed after the designation by the UNESCO Program."

To strengthen the regional network, in January 2020, the MHBR Council for the first time created the "MHBR Community Development Exchange Workshop", where local residents from four prefectures and seven cities surrounding the MHBR came together to discuss common issues and future actions towards sustainability in the Circum-Hakusan Area [62]. The topic of the meeting was "Connect and Expand the Harmony of Hakusan" and the objectives of the meeting were constructed according to the LAP and MAB Strategy:

- To create a platform for dialog and sharing information about local issues;
- To revive the area through the creation of new values and increasing the attractiveness of the region;
- To provide recommendations for future actions.

The outcomes of the meeting are compiled in the Table 7.

**Table 7.** Outcomes of the first "Community Development Exchange Meeting" of MHBR members. The most common replies are ranked from highest to lowest. Ranking order shown in brackets.

| Common Questions |
| --- |
| - *How can communities survive with depopulation and no followers? (1)* |
| - *What kind of support is expected from the Government? (2)* |
| - *In what direction is Tourism promoted? (3)* |
| - *How do we deal with Invasive Species? (4)* |
| - *How do we sustain the decreased number of local hunters, the management of traditional houses and cultural assets? (5)* |
| - *How do we preserve the decrease of alpine plants due to increased tourism and climbing? (6)* |
| - *How do we link the activities of every resident with MHBR? (7)* |
| **Common Issues:** |
| - *Lack of Human resources to sustain livelihood (1)* |
| - *Less chances to learn about the activities of MHBR, and a lack of awareness by local residents (2)* |
| - *Difficulties to live as a family inside BRs due to separated work/study stations of family members (3)* |
| - *No network between other regions (4)* |
| - *Less youth involvement during local events and festivals (5)* |
| - *Personal activities not linked with the activities of MHBR (6)* |
| - *No access to information about the activities inside the region (7)* |
| **Recommendations:** |
| - *Nominate a Coordinator who will CONNECT and COMMUNICATE between all four prefectures and local communities inside MHBR (1)* |
| - *Create an education platform to educate and share knowledge with locals (2)* |
| - *Create a concrete plan for the MHBR Project with the involvement of local residents (3)* |
| - *Increase the number of meetings between the locals in MHBR (4)* |
| - *Increase human connection and cooperation (5)* |
| - *Create a region with new job opportunities to attract the youth (6)* |

For MHBR locals, issues such as depopulation, lack of awareness, as well as networking and communication with local authorities of BRs were the main problems. It was clear that the locals were not engaged with local officials in the participatory processes of managing BRs.

A local guide state, "Most of the time, each community is conducting its own activities without informing others, and we lack communication between communities. Also, we are not informed about the projects of the government, and we know about them only after they are implemented."

### 4.2. Role of Local Communities in KatBR

Local communities of the KatBR have been developing a balanced interaction with the environment and surrounding nature for many years. However, during the economic crisis in the 1990s and the collapse of the Soviet Union, which resulted in high unemployment, many local people lost their basic sources of income. In order to survive, they turned to

illegal hunting and collecting medicinal plants and other non-timber forest products. This resulted in significant threats to biodiversity loss within the BR. A dramatic decline in the Musk deer population has been observed since the 1990s due to overhunting [63,64]. A Ust-Koksa villager stated, " . . . our main issue is the lack of constant job opportunities. During the Soviet period, most of the villagers were working in factories and had a stable income, but now Nature has become the only source of income. Therefore, to sustain families, some people started illegally hunting Musk Deer, and selling it on the black market."

With such obstacles, the KatBR faced a situation where the implementation of conservation function was directly affected by the economic status of local communities. Therefore, it was clear that to conserve the environment there should be alternative sources of income for local people. For this reason, in 2012, the KatBR administration together with the local NGO, "Altai-Sayan Mountain Partnership", started a program for the development of ecological and rural tourism in the BR and its surrounding area [65]. The aim of this program was to provide alternative sources of income for local communities and to establish a situation where local people became economically interested in nature conservation (rather than in its overexploitation). It was obvious that promoting ecotourism would provide such opportunities for locals. At its initial stages, the program included three basic components:

1. The financial support of small environmentally friendly local businesses by means of interest-free microcredits distributed among people living in the BR on a competitive basis.
2. The education of local people on various aspects of running a business.
3. The assisted promotion of local goods and services through a special website, participation in different fairs, publishing and dissemination of booklets, catalogues, etc.

In recent years, the program has covered not only issues of tourism, but also other economic aspects as well, primarily the production of local handicrafts and food such as honey and milk products.

A local authority stated, "The designation of the territory as a UNESCO Biosphere Reserve gave us more international recognition and increased the possibility to apply for the external funding and to conduct international projects. This designation helped change people's minds about the importance of nature protection for the stable economic development of the region."

Nowadays, this vision has evolved from "the support of individual households" towards "the complex development of entire villages". The public council provided a platform to discuss the issues of local development with local communities and how to reconcile the development with conservation and natural and cultural values of the territory. The vision was a brand-new paradigm of nature- and culture-based tourism called "slow travel" [66]. Unlike "fast travel" in a big group with a non-local guide, when tourists try to see more attractions during their short visit, the "slow travel" model of journey [67] supposes tourists' deep cognitive cultural immersion in a local environment combined with high ecological standards and maximum participation of local communities. In the framework of this model and cultural exchange, tourists receive impressions and emotions which could serve as a trigger for further personal transformation towards a newly realized attitude towards the environment. According to the LAP and the MAB Strategy, along with participatory approach, the overall goal of the KatBR is considered to be "to change the attitudes of people towards sustainable consumption and to balance interaction with nature".

Up until now, this program resulted in the following output in the KatBR and its surrounding area:

- Local communities developed 310 business-plans ready to be implemented.
- 185 families received direct financial support from the KatBR. Jobs were created by the local villagers.

- More than 600 new jobs were created (including 150 all-year).
- Through training, 150 people received education (and relevant university-issued diplomas) in various aspects of tourism.

To understand the program's impact, the KatBR monitored the social acceptance of the BR. From 2012–2020, the level of public support of the BR increased from 68% to 93%. One of the most significant outcomes of this program was the elimination of illegal hunting in the core and buffer zones of the BR (in 2010, during antipoaching activities, the KatBR demounted more than 100 illegal snares for the Musk deer per season, while in 2019–2020, there we no snares installed in the core and buffer zones).

In addition, the KatBR provided opportunities for the local villagers to obtain diplomas and participate in additional training courses called "Management of Rural Tourism" and "Tourist Guiding and Instruction for Horseback Tours" developed together with the local Gorno-Altaisk State University. Annually, 20–30 people living in the KatBR participate in such training. Many local villagers work as local guides, and this kind of diploma helps to increase their sense of responsibility towards the protection of nature.

A local guide state, "I was unemployed for many years . . . I came to KatBR and the authorities helped me to find a job inside the BR. Now, I work as a local guide for tourists and as a driver... My perception towards nature conservation changed very strongly after I started working with KatBR. I feel a responsibility to protect Nature, and I feel more responsible for my own actions."

## 5. Discussion

The interviews demonstrated that Japanese and Russian communities who live inside the protected territories have different attitudes towards the BRs. The involvement of locals in regional development strongly depends on how governmental agencies manage the participation of locals inside BRs. Different levels of participation described in the frameworks were found in both BRs and varied for different activities. According to the interview data from the authorities and local communities living in the periphery of each MHBR and KatBR, we found that the differences between local Japanese and Russian engagement in the regional development of BRs were derived from three main reasons: (1) historical relationships between the governments and the local communities, (2) attitudes towards the concept of BRs and their economic benefits, and (3) perceptions of nature protection. Below, we discuss those categories in more detail.

### 5.1. Historical Relationships between the Government and the Local Communities

One of the main differences between the KatBR and the MHBR was the level of participation due to government systems. In the case of the KatBR, the local government is trying to provide opportunities for local communities to receive economic development and education opportunities by using BRs, i.e., engage in the activity-specific and active participation (level D and E). Whereas, in the MHBR, for years, local people have had their income resources inside designated areas due to the land ownership system, and they do not rely on help from the local government to increase economic conditions. This kind of independence was the reason for the nominal or passive participation (level A and B) in MHBR for many years.

The results showed that the main issue of the KatBR officials was to ensure a sustainable budget and funding system to stop communities from illegal poaching and land use. Whereas, for the MHBR officials, the lack of human resources and issues with depopulation and land abandonment were the main problems. Historically, until the 1990s, the Russian system of nature management with the strict state-run nature protection regulations, "Zapovednik", was funded by the government and was mainly research oriented. However, after the fall of Soviet Union in 1991, the economic, political, and social systems in Russia were totally reorganized. As a result, 60–80% of federal funding was reduced [68], forcing Zapovedniks to search for new funding sources. Budget cuts posed the greatest threat to manage Zapovedniks and BRs. They had no funds for scientific research and lacked basic

infrastructure and equipment to patrol the protected areas [69]. In addition, the lack of law enforcement combined with poverty led to the escalation of illegal poaching, logging, and mining within the protected areas [70]. In 1995, a new law on "Specially Protected Natural Areas" was introduced and played a significant role in transforming Russian nature management strategies. For nature protection, research, and monitoring, one new goal, "environmental education", was added as well [71]. Section 2, Article 7 describes the development of "environmental education and environmental tourism", which allowed Zapovedniks and BRs to be accessible to the general public and added new sources of income by developing eco- and recreational tourism [9,72]. Communities and local villagers became the main actors to contribute to fundraising (interactive participation level F), and tourism helped to increase employment opportunities inside the BRs. The KatBR was one of the first BRs in Russia who took the initiative to develop educational programs for local communities. In order to provide an income for local people, the officials of the KatBR, with the support of WWF, implemented educational and training programs, in which they launched a program involving the local people in the development of ecotourism inside the BR. Later, it was changed to the development of local business such as production of local handicrafts, food products, house renting, etc. The officials of the KatBR constantly organize programs by teaching about the establishment and management of a small business, the management of camping houses, tourism guide trainings, the production of local goods, etc. This kind of support has helped local people to feel that they are not only business owners, but they have a responsibility to protect nature. The shift from top-down governmental management towards participatory approaches (levels D, E, and F) by the local communities has helped the BRs to conduct a sustainable management of protected territories. It has also helped to promote new models of citizen participation and increase the opportunity for self-funding to conduct research and build the relevant infrastructure.

The Japanese approach to the involvement of local communities in nature conservation inside BRs is different from the Russian approach. This difference comes from the specificity of the Japanese nature conservation system. Japan has a unique park management system and has never had a nature protection system as strict as the Russian Zapovedniks. The most renowned and extensive system of nature conservation in Japan is the system of natural parks which consist of national, quasi-national and prefectural parks [73]. The management system of national parks is called "*Chiiki-sei*", which is distinguished by its multiple-use system and zoning management. The most prominent features of Japanese national parks are that, unlike Russian parks, the parks are not established in the state-owned land, and the land is not "set aside" for nature conservation [74,75]. Instead, the areas are designated as parks by the Ministry of Environment (MoE), often by the request of local people and communities, for their "scenic beauty" and necessity to preserve the landscape. People continue to live and engage in land management activities and all socio-economic activities of the local communities and stakeholders take place within the regulation rules [76]. However, even though the national parks areas are regulated by the MoE, the management of BRs belongs to the Ministry of Education, Culture, Sports, Science and Technology (MEXT), which protects natural resources under the Law for the Protection of Cultural Properties. The complexity of this governmental management system in these protected areas creates significant issues and a lack of coordination. Top-down approaches towards the nature conservation provides very few opportunities for local communities to participate, so their opinions are rarely reflected in the decision-making processes of BRs (levels A–C). From 2012, after the adoption the Seville Strategy, the concept of BRs in Japan started to change from "research-oriented" to "community-oriented" (levels D and E). This change started to attract more local actors to utilize local natural resources for sustainable regional development [77,78].

The MHBR has shown a case where, for more than 30 years, local people did not show any interest in the program. They were continuing their activities and livelihood without acknowledging that the area was designated by UNESCO's Man and Biosphere Program. This was because the designation of BRs was already on the same territory of

a national park. Giving a new name to the protected area made no changes. This was also confirmed by a local's statement that " ... we have been living in this territory for many years and nothing changed after the designation into the UNESCO Program." The new designation did not create any significant changes for locals. In addition, top-down implementation and multi-governance of the MHBR complicated the involvement of local people in the decision-making processes. Recently, levels E and F have been observed on a small scale, but still level G is not observed at all in the MHBR. Some other studies also showed that a failure of a great number of BRs sites was because the BR nomination was only "rebranding" the existing designations of national parks [78,79]. Due to the limited communication and low public support, the MAB Program was less popular than other programs such as the World Heritage and National Parks [80].

*5.2. Attitudes towards the Concept of BRs and Economic Prospects*

The people's attitudes towards the concept of BRs in both countries varied significantly. We observed that the varying knowledge of economic prospects and benefits from BRs was one of the reasons why local communities acted differently in each case study. Here, we assume that ecocentric and anthropocentric attitudes towards nature protection [39] can create those differences. The locals of the KatBR showed more of an anthropocentric approach to nature protection. Local people value and protect nature for its positive effects on their lives and consider nature as the main source of their incomes.

One local stated, " ... when we had no job to sustain our lives, Nature was feeding us, and helped us survive during the hard 1990s ... ." One of the basic principles of the Altaian philosophy of life was "alysh-berish" (literally "to take = to give"), i.e., Mother Earth provides different benefits, and if people use those benefits, they need to give back with gratitude and worship. The overall principle of using ecosystem services is proportion, meaning that one should not take more than is required to support one's livelihood. The Russian Old-Believers had the same philosophy combined with the ethics of labor and isolated lifestyles. This paradigm has been more or less affected by the social transformations during the 20th century, but it is still conserved as (at least) the cultural heritage of Altaians and Old-Believers. It was clearly observed in the KatBR that the designation of BRs provided opportunities for economic development through the promotion of ecotourism and small businesses. According to the monitoring results, the BR designation shifted the attitudes of the locals towards sustainable consumption and balanced human–nature interactions. In addition, constant training and educational activities inside KatBR helped raise awareness and changed people's stances on the region's sustainable development. Historically, Russian "Zapovedniks" put restrictions on human activities to preserve the flora and fauna from overuse and prevent the exploitation of natural resources. Strictly regulated by the government, local people had no access to those territories except the specially designated areas where recreational and educational activities were allowed, but with no economic activities. Adding the title "Biosphere Reserve", or sometimes locally called "Biosphere Zapovednik", along with changes in the legislative system, made the concept more accessible for locals and created possibilities for international cooperation [81] and more employment opportunities. KatBR officials used the participatory approach (levels E and F) and developed 310 business plans and created more than 600 new jobs.

The Japanese attitude towards nature is more ecocentric as they do not perceive nature as a main source for economic development. Japanese people appreciate nature mainly for its *"scenic beauty"* [73]. Japan has a long history of landscape management through human–nature interactions. For national parks, local authorities cannot exclude the involvement of local communities and other stakeholder in land management and park utilization due to the landownership system. Even the designation of national parks in landowners' territories had little value, because they had nothing to lose. They were the main landowners and continued to live and conduct their daily livelihood activities [73]. Local authorities cannot put any restrictions on local communities concerning access to

natural resources or conduct community relocations for nature protection, as could often happen in Russian Zapovedniks. There were no restrictions, relocations, or detriments to the lives of Japanese landowners. Therefore, the title of national park was easily accepted and welcomed by local communities as it only had benefits. It increased the region's value for tourism development and possibilities for the economic growth of the region. However, the supplemental designation of national parks also as UNESCO's BRs did not make any significant changes for locals. This lack of recognition by the locals was caused not only by the complicated governance system, but also by the lack of awareness of the benefits and opportunities for economic development after the designation.

Additionally, during the first community workshop conducted by the MHBR Council, locals of the MHBR strongly stated they had a lack of knowledge and awareness of the concept of BRs. The workshop participants emphasized the importance of educational activities to share knowledge and skills between locals, and how it could be possible to raise awareness on the concept of BR to deal with the lack of job opportunities and depopulation. One of the main recommendations was to create a platform to "connect and communicate" with the local communities of all four prefectures (Table 7). Here we can assume that the attitudes towards the BR can be changed according to how well local people are aware of its concept. These results are supported by the studies of Van Cuong et al. [26], who also highlighted that adequate "awareness and communication" (Table 1) leads to willingness to support BRs and promotes participatory activities. Van Cuong et al. also argued that the effective participation inside BRs required "information sharing and communication" through many multi-directional approaches that provided equitable knowledge-sharing opportunities and well-structured dialogue among participants.

*5.3. Perception of Nature Protection*

We can also assume the spiritual aspect of nature protection creates different attitudes towards the protected territories of BRs. The concept of nature protection was not popular in Japan because nature was viewed as intangible. On the one hand, instead of believing that nature was a destructible substance that needed protection, they spiritually appreciated it as a deity that was always accessible. Japanese people had a spiritual connection with the natural environment (*shizen*) rooted from Buddhist and Shinto beliefs [82]. Hilly mountains areas were places where *Kami* (Deities) reside, and no human activities were allowed there. Natural features such as mountains, forests, trees, caves, waterfalls, and hot springs were worshiped as sacred places for deities [83]. Worshiping nature rather than protecting it was the main attitude of Japanese people. Even though Western influence started to dominate at the beginning of the 20th century and the concept of the national park was introduced from the Western world, local people have still kept the spiritual connection with nature and continue nature appreciation in a spiritual way.

Russian people, on the other hand, perceived nature as a living "Mother Nature", who provides the basis for all productive forces, sustainable economy, and livelihoods. They felt a strong necessity for nature protection and "wise use" [84]. Nature was appreciated for its beautiful landscapes and aesthetic values and was always described this way in Russian literature. This prevalence of nature description in Russian literature also has a connection with starting Japanese nature preservation. The influence of Russian literature on forest landscape appreciation in Japan, such as with deciduous trees, is one example of this connection. After reading poetic descriptions of forests by Russian writers such as Ivan Turgenev and Leo Tolstoy, Japanese people were encouraged to consider the beauty of deciduous forests on their own. This can be considered as the first step towards the preservation of forest landscapes in Japan [42,85].

## 6. Conclusions

This study showed that the involvement of local communities in regional development depends on many aspects such as the country's history of nature protection, governmental management systems, economic conditions, land ownership systems, and cultural

perceptions of the locals. In the two case studies, access to decision-making processes and knowledge affected the participation levels of the citizens. The locals in the KatBR function within a participatory-based system (participation levels E and F) having a greater role in decision-making processes. Some of their abilities to participate is facilitated by government management systems (e.g., The KatBR Public Council shown in Table 5). In the case of the MHBR, low participation and a lack of knowledge about the benefits of the BR designation are still the predominant factors decreasing the participation level of the locals. The MHBR lacks a council or department to educate locals about the benefits of the BR. Including a Public Council or Department of Eco-Education as seen in the KatBR could allow BR residents to gain motivation and increase their level of participation to reach levels E, F, or G. Increasing these participation levels is an essential goal for all BRs, and the differences found between these two cases can also be applied to other cases. More interaction between officials and locals, along with greater education about the economic benefits of the BR, play key roles in their functioning. Since no level G participation (self-mobilization and taking over of the responsibility) was seen in either of the two cases, future studies should be conducted to examine how to further increase participation levels by BR locals. These results can contribute to the MAB strategies and WNBR, which emphasize effective collaboration and share lessons learned for a better governance and management of BRs. Partnerships within the network can also contribute to achieving their goals in a sustainable manner, resolving common issues and promoting nature management.

The study also found differences in both case studies between the designations of "national park" and "biosphere reserve" and how the land ownership systems of each country were related to these designations. Future studies must be conducted to explore and analyze this relationship and its effect on participation in BR management.

Points that we did not address in this study were the differences in structural formation of the communities, socio-economic status, knowledge and educational background, and the role of each stakeholder in the management of protected areas. More analysis related to the community formation and structure must be conducted in future studies to examine these differences between Russian and Japanese BRs.

**Author Contributions:** A.M. conducted the principal investigation inside both counties, conducted the interviews for local communities, officials, and NPOs, and after the analysis, wrote and drafted the entire article; C.D.S. revised and edited the manuscript along with consulting the authors about the manuscript's revised structure and contents; T.Y. provided the materials about Russian Katunskiy Biosphere Reserves, wrote some parts related to KatBR, and provided suggestions for the entire manuscript. All authors have read and agreed to the published version of the manuscript.

**Funding:** This study was financed by the Japanese Grants-in-Aid for Scientific Research—KAKENHI.

**Institutional Review Board Statement:** Ethical review and approval were waived for this study, due to reason, that this study made no evaluation analysis on humans, and all collected data were publicly available.

**Informed Consent Statement:** Informed consent was obtained from all subjects involved in the study.

**Data Availability Statement:** All data is stored at Kanazawa University, and is available from the corresponding author on reasonable request.

**Conflicts of Interest:** The authors declare that they have no competing interest.

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
