# Peer review of "Comparative Analysis between the Role of Local Communities in Regional Development inside Japanese and Russian UNESCO’s Biosphere Reserves: Case Studies of Mount Hakusan and Katunskiy Biosphere Reserves"

_sustainability, doi:10.3390/su131810422_

Round 1

Reviewer 1 Report

This paper contains interesting information about the role of local communities in managing Mount Hakusan Biosphere Reserve in Japan and the Katunskiy Biosphere Reserve in Russia .    Findings  showed that the involvement of local communities in regional development depends on many aspects such as  the  country’s history  of nature protection, governmental management systems, economic conditions, land ownership systems and cultural perceptions of the locals. 

Author Response

Dear Reviewer 1, 

Thank you very much for your high evaluation of our Mansucript. 

Reviewer 2 Report

The manuscript has been seriously corrected, but there still remains some shortcomings:

A title of the manuscript is too long, I suggest remove the second sentence from the title.

Structure of the manuscript must be changed: Sections no. 1 and no. 2 must be join to one section entitled “Introduction”. Also, there is still too much redundant text in lines 99-246, and it must be seriously shortened. Figure 1 is redundant and should be removed. Figures 2 and 3 have very different quality (both are very poor and redundant – not important for analyse and results), I suggest remove both figures. Tables 1 and 2 are “adapted” from another source and must be removed (it seems to be a plagiarism), BTW it is enough to use relevant citations in the text beyond these tables.

Subheadings in line 248 and line 432 need improvement (Authors should use only “Methods and Material; Results”).

Authors must better clarify, what methods exactly were used (line 267), Really, there was used any “comparative analysis”? What it means exactly? Why Authors did not used any statistics in validation of qualitative data?

Section Results must be seriously corrected, and text looks like “discussion” must be removed (see e. g. line 528).

Section Conclusion should be checked – authors should use more professional words (e.g. not “showed” in line 779, but better “indicated” etc. authors also should add to the end of the Conclusion section a brief statement how results of the study are interesting or important (1) in international scale and (2) for concept of Biosphere Reserves.

Author Response

Please find replies in the attached file.

This manuscript is a resubmission of an earlier submission. The following is a list of the peer review reports and author responses from that submission.

Round 1

Reviewer 1 Report

Dear Authors, please find my comments bellow:

Rating the Manuscript Originality/Novelty: Comparative Analysis of such different Systems - is not
really a significant novelty. Conclusions for the KAZ/RUSS case study are
appropriate and interesting for current discussion for a real community
based implementation of the Seville Strategy on BR = Focus of my comments Quality of Presentation is good - introduction of BR history could be shorter
- as for my point of view Scientific Soundness is appropriate Overall Merit: Paper provides current knowledge on participatory
management of BR - better to focus only on history since "Sevilla" - Overall history of BR development bit to detailed - Current changes since "Seville Strategy” for Biosphere Reserves should
be focus - and expansion of the BRs with transition zones to promote regional
sustainable development plus utilization of natural resources by the locals
could be more detailed. Main concern on BR-Strategy is how can be increased the level of public
support of the BR - My institute introduced the approach of action research
and implemenation of concrete action on the ground - in a research triangle
University as researcher - NGO and local government and PA administration
(see my comments in pdf)

Reviewer 2 Report

General remarks:

This article discusses the role of local communities in regional development issues within the context of UNESCO’s Man and Biosphere Programme, using two Biosphere Reserve cases: Mt. Hakusan, Japan (HAK) and Katunskiy, Russia (KAT). The aim was to compare these two examples, as HAK follows a top-down management approach with less local engagement, whereas KAT shows a bottom-up approach with intensive local participation. The paper postulates that the differences in local engagement in the management of BRs depend on the cooperation between governments and local institutions, the attitudes of the locals towards their BRs, and the historical perception of nature protection.

Critical comments in detail:

  • 29: Please amend: It was in 1976 when the first BRs (57 in number) were founded.
  • 74f.: Other examples with similar problems, and heaps of literature dealing with what you find worldwide: huge BR perimeters (e.g. sub-Saharan Africa); demographic change (e.g. Europe with Switzerland, Germany, France etc.).
  • 81ff.: The statement “Until now, researchers in BRs were focused mainly on biodiversity monitoring…” is incorrect. During especially the last 20 years international scholars have published plenty of papers on BR topics, including participation, governance, and park-people relationships. We just analyzed the situation of community involvement in all 44 Mexican BRs and found 121 peer-reviewed articles published since 1981.
  • 145f.: Please amend: The first UNESCO conference was held in 1983 in Minsk, Russia … (it is your duty to elaborate on the details).
  • Chapter 2: two simple maps of the Japanese and Russian BR systems would make a lot of sense.
  • Before Chapter 3 this paper misses two more chapters: firstly, a real state-of-the-art literature review, and secondly, a theoretical framework (e.g. political ecology, socio-ecological systems framework, (regional) governance).
  • 216f.: Here it is necessary to elaborate on your executed methodologies: did you do a systematic literature review, then you need to mention the number and specifications of the sources; when and how long did you do fieldwork; with whom (stakeholder positions) did you talk, when, and for how long. You also need to clearly differentiate between what you mean with “1. General aspects of BRs” and “3. Issues of BRs”?
  • 234-347: Try to make another table out of these 4 points (like your table 1).
  • 254ff: this part reads very idiographic.
  • Figure 1: an inset map of Japan would be fine to show where this BR is located; ditto for figure 2 – the style of these maps might also be adjusted.
  • 339: How many ecotourists/visitors does the BR receive per annum and where do they come from?
  • Table 1: Please amend the percentages of the three different zones; furthermore, the number of municipalities for KAT is missing.
  • 444: Relating to “Slow travel” you find an unscientific source under: https://en.wikipedia.org/wiki/Slow_Tourism
  • 457ff.: Didn’t the 185 families get support from the BR? What kind of jobs has been created, by whom?
  • 461ff: Please add quotes for the rising public support, illegal snares, training participants, and the role of tourism (here some figures are also missing).

Overall scientific evaluation:

This paper raises an interesting and very much en vogue topic. Unfortunately, this contribution is not a scientific one. Relating to the latter, three major shortfalls need to be addressed:

  1. the article is missing a theoretical frame;
  2. there is no state-of-the-art literature review;
  3. if this is supposed to be a qualitative empirical analysis, the evidence with direct quotes of experts interviewed is missing.

Hence, my strong plead is for ‘rejection’.

Reviewer 3 Report

This paper contains some interesting information on the two BRs in and contrasting socio-economic contexts. It tackles some interesting research questions and provides some, although rather shallow, results. Overall, scientific rigour is largely missing, i.e. the research questions are completely general and do not correspond to what is being found after, it is mostly unclear where the information that is provided in the text is from and how it was used to write the text: with whom were interviews made, what was asked, how were the responses analysed, how were they fitted together to get a text. In the discussion section new aspects are being introduced, combined with some new results. Discussion, comparisons and explanations with other papers are rather shallow.

To me, this is an interesting text for a non-scientific journal, as it contains many interesting and nicely written parts. However, for a scientific journal, it needs a drastic revision.

Reviewer 4 Report

- It is necessary to make two units - Introduction and Literature review with special emphasis on the role of the local community as one of the key stakeholders in the development of the destination (seek recent research on this topic, ).

- Paint all corrections in red color

Reviewer 5 Report

This article sets to compare the role of local communities in managing BR in Japan and Russia. The authors show their in-depth knowledge of the system of Protected Areas in these two countries and promise to demonstrate a meaningful comparison which will be able to provide lesson learnt for management of BR in other contexts. However, the article, in my opinion, is in need of substantial revision.

The foremost shortcoming of this article is the weakness in its argument. The authors argue that KatBR is based on “bottom-up” approach because the management involves local communities. In the article there was no evidence showing how the local communities are involved in “decision-making” process. Note that “bottom-up” approach does not simply means government creating job/economic opportunities or help local people. These can be just superficial “local participation”. See the following reference for an analysis of “local participation” in protected areas:  Wang J (2019) National parks in China: Parks for people or for the nation? Land Use Policy, 81, pp. 825-833, DOI:10.1016/j.landusepol.2018.10.034. A genuine bottom-up approach in NRM means local community is the main actor of making management decision. There was no evidence demonstrated in this article which provide strong argument for this. I think this is the single biggest problem of this article. The authors need to provide more information, for example, how the local NGO mobilize local community to participate, etc.

A second problematic argument is the conflicting statement in its Discussion (5.2). The author argue that Japan “ has a long history of landscape management through human-nature interaction, and the authority cannot exclude local stakeholders…”. If that’s the case, why do the author also argue that Japanese BR has less community involvement?

5.3 is also problematic. If Japanese has a culture to respect Nature and deities, how is it possible for them not to protect but to destroy the Nature? The argument is very confusing.

Moreover, the author need to elaborate why these two BRs are chosen, as they are very different in terms of population density, local livelihood etc. The reasons stated in the article were insufficient.

In terms of writing, there are also some room to improve. The authors are encouraged to seek professional editing service for improving English. Also, a map showing the locations of the two BRs in relation to the countries (not just the BRs themselves) is needed in the very early part of the article, for readers to familiarise the context.

The author should detail the research methods, such as detailing how many interviews, what workshop attended etc.

The Introduction should include a roadmap of this article, showing the structure of the article.

Finally, the conclusion should also be revised. This is a qualitative research so the conclusion should be based on the solid finding. For example, what are the lessons to be learnt for future initiative etc. Another quantitative study is not necessary needed.

Overall, there are some positive points in this article but the logic of the argument need to be strengthen significantly.

Reviewer 6 Report

Generally, the manuscript is poor, and it is not well written, but I have detected a potential for major improvement of it. Basically, English language is not very correct and needs major improvements (see using capital letters in common terms such as in line 19 etc.).

Authors should better use “scientific writing style” (e. g. in line 11 not “we decided”, but “The topic of this study is aimed to…”; in line 14 not “we choose…” but “based on case studies in Japan and Russia”; in line 17 not “we have suggested…” but “Results of this study indicated…” etc.). Also using abbreviations in the section Abstract is not appropriate.

Line 11 – There is missing brief wider background of the topic of study (max two sentences).

Lines 20-21 – Why Authors did not do these necessary more quantitative analyses? By the way, I think there should be correctly “more qualitative analysis”.

Line 22 – There is missing clear statements related to main original results of the study, which can be interesting for international audience of the journal (see also my comments to the section Conclusion below).

Line 56 – Add here relevant citations (Oprsal, Z. et al. What Factors can Influence the Expansion of Protected Areas around the World in the Context of International Environmental and Development Goals? Probl. Ekorozw. 2018, 13, 145–157.).

Lines 214-128 – Aims(s) of the study must be reworked in accordance to the title and Abstract (line 13). What is exactly the aim of this study? If it is comparative analysis (not only description), Authors must add here any hypotheses in tested or clear research questions.

Line 129 – 214 - Section number 2. (Overview…) is redundant (see the same information in lines 248-352) and must be removed. Some minor and really important information (clearly related to the topic of the study) from this text must be moved to the sections Introduction and Methods.

Line 215 etc – Section Methods is very poor. There is too much redundant information, and on the other hand, there are missing basic information related to methodology of research (how many respondents were investigated, more information related to respondents are missing, questionnaire must be included to the appendix of the study, what statistical methods were used to analyse of interviews etc.). Text in line s 221-230 is confused without any scientific value. Section Methods must be completely improved to serious clear and scientifically correct form.

Lines 248-352 – Very long and mostly redundant text without clear connection to the defined aim of the study.

Lines 352-472 – This section is a mix of results and methodological text. E.g. in lines 393-397, this text is “Methods”, not “Results”. This section must be reworked following basic rules of scientific writing.

Line 473 – section Discussion seems to be interesting and can be valuable part of the manuscript. But, the structure of this section should better follow the structure of the manuscript (defined research questions/hypotheses and main original results). It can be reworked and addition of more international literature related to protected areas seems to be possible (see e.g. discussion in the study doi: 10.3897/natureconservation.24.21608).

Line 609 – section Conclusion is too general. Authors should present here main scientific original results of the study, and (based on previous Discussion) highlighted international relevance and portability of results. It is especially important in the topic of biosphere reserves concept, which is world-wide.

Line 634 etc: Reference do not follow rules of the MDPI journals and must be corrected (and extended, see my comments above).